# Ethanol Production from Wheat Straw Hydrolysate by Issatchenkia Orientalis Isolated from Waste Cooking Oil

**DOI:** 10.3390/jof7020121

**Published:** 2021-02-06

**Authors:** Alexander Zwirzitz, Lauren Alteio, Daniel Sulzenbacher, Michael Atanasoff, Manuel Selg

**Affiliations:** 1Biosciences Research Group, University of Applied Sciences Upper Austria, Stelzhamerstraße 23, 4600 Wels, Austria; daniel.sulzenbacher@gmail.com (D.S.); Michael.Atanasoff-Kardjalieff@students.fh-wels.at (M.A.); Manuel.Selg@fh-wels.at (M.S.); 2Centre of Microbiology and Environmental Systems Science, Department of Microbiology and Ecosystem Science, Division of Terrestrial Ecosystem Research, University of Vienna, Althanstrasse 14, 1090 Vienna, Austria; lauren.alteio@univie.ac.at

**Keywords:** *Issatchenkia orientalis*, cellulosic ethanol, wheat straw, biorefinery

## Abstract

The interest in using non-conventional yeasts to produce value-added compounds from low cost substrates, such as lignocellulosic materials, has increased in recent years. Setting out to discover novel microbial strains that can be used in biorefineries, an *Issatchenkia orientalis* strain was isolated from waste cooking oil (WCO) and its capability to produce ethanol from wheat straw hydrolysate (WSHL) was analyzed. As with previously isolated *I. orientalis* strains, WCO-*isolated I. orientalis* KJ27-7 is thermotolerant. It grows well at elevated temperatures up to 42 °C. Furthermore, spot drop tests showed that it is tolerant to various chemical fermentation inhibitors that are derived from the pre-treatment of lignocellulosic materials. *I. orientalis* KJ27-7 is particularly tolerant to acetic acid (up to 75 mM) and tolerates 10 mM formic acid, 5 mM furfural and 10 mM hydroxymethylfurfural. Important for biotechnological cellulosic ethanol production, *I. orientalis* KJ27-7 grows well on plates containing up to 10% ethanol and media containing up to 90% WSHL. As observed in shake flask fermentations, the specific ethanol productivity correlates with WSHL concentrations. In 90% WSHL media, *I. orientalis* KJ27-7 produced 10.3 g L^−1^ ethanol within 24 h. This corresponds to a product yield of 0.50 g g^−1^ glucose (97% of the theoretical maximum) and a volumetric productivity of 0.43 g L^−1^ h^−1^. Therefore, *I. orientalis* KJ27-7 is an efficient producer of lignocellulosic ethanol from WSHL.

## 1. Introduction

Production of fuels from sustainable sources has been a widely investigated topic during the last two decades. Particularly, much research was focused on the biotechnological production of cellulosic ethanol from agriculture-derived lignocellulosic residues, like corncobs, sugar cane bagasse or various kinds of straw. This conversion of lignocellulose-derived sugars into ethanol covers a range of individual processes: pretreatment of the material, enzymatic hydrolysis, fermentation and downstream processes [1]. The fermentation process is strongly influenced by inhibitory substances that are generated during the pretreatment of the lignocellulosic substrates [2]. Hence, microorganisms that are more tolerant than others to inhibitory substances, like carboxylic acids and phenolic compounds, are favored for biorefinery applications [3]. *Issatchenkia orientalis,* also known as *Pichia kudriavzevii*, *Candida krusei* or *Candida glycerinogenes* [4], can be readily isolated from food and environmental sources and offers tolerance to various stresses like low pH, high salt and high temperature [5,6,7,8,9,10]. Hence, *I. orientalis*, which is also involved in wine fermentation [11,12,13], has been utilized to produce various value-added compounds and platform chemicals under highly acidic conditions [14,15,16,17,18]. Due to its outspoken tolerability to various stresses, *I. orientalis* has been considered a promising alternative for sustainable ethanol production [19,20]. Furthermore, it offers tolerance to classical fermentation inhibitors that are generated during pre-treatment of lignocellulosic substrates [21]. Therefore, *I. orientalis* has been used for the production of cellulosic ethanol from low cost substrates that are derived from acid-impregnated and subsequently steam exploded lignocellulosic materials like plywood chips, rice straw and sugarcane bagasse [22]. In addition, *I. orientalis* strains that grow particularly well on lipid substrates and are able to accumulate large amount of fatty acids have been described [23,24,25]. Likewise, an oleaginous *I. orientalis* strain isolated from rotten fruits, has been used for the production of biodiesel [26]. However, compared to conventional and other, more renowned, non-conventional yeasts, the amount of available literature on the engagement of *I. orientalis* in biorefinery processes is rather limited. Nonetheless, an increasing interest in this microorganism is demonstrated by the facts that the genome of such a highly tolerant *I. orientalis* strain has recently been sequenced [27] and that tools for genetic engineering of *I. orientalis* have been developed [28,29].

The objective of the present study was to discover novel yeast strains that can grow on wheat straw and ferment it to ethanol. Generally, oleaginous yeasts have been employed for the biotechnological valorization of lipid containing waste streams, like waste cooking oil (WCO) [30,31,32,33,34]. Plausibly, biotechnologically interesting organisms that up-cycle WCO have been isolated from WCO [35]. In the present study, we describe the isolation of an *I. orientalis* strain from WCO, its growth characteristics and its capability to efficiently produce ethanol from steam exploded wheat straw hydrolysate (WSHL).

## 2. Materials and Methods

### 2.1. Isolation and Identification of I. orientalis from Waste Cooking Oil

Waste cooking oil samples were collected from a local restaurant. Samples were taken with a sterile pipette from 30 cm depth. 100 µL of the samples were transferred to plates with Rose Bengal Red agar containing chloramphenicol (RBC) from Carl Roth, Karlsruhe, Germany. To obtain pure cultures, single colonies were transferred to a fresh YPD (Carl Roth, Karlsruhe, Germany) plate containing 15 g L^−1^ agar (VWR, Vienna, Austria) after 3 days and were again incubated for 2–3 days. The strain was maintained on YNB (Yeast Nitrogen Base without amino acids 6.7 g L^−1^ from Carl Roth, Karlsruhe, Germany) plates containing 20 g L^−1^ glucose (YNB-D) and 15 g L^−1^ agar. From these pure cultures, genomic DNA (gDNA) was isolated as described by Looke et al. [36] and used as a template for PCR amplification of the internal transcribed spacer (ITS) 1 and 2 regions including the 5.8S locus of the ribosomal DNA (rDNA). The primers used for PCR amplification were ITS4 (TCCTCCGCTTATTGATATGC), which binds at the 28S large subunit rDNA locus, and ITS5 (GGAAGTAAAAGTCGTAACAAGG), which binds at the 18S small subunit rDNA locus. The use of these primers for identification of fungi and yeasts has already been described elsewhere [37]. The approx. 550 bp PCR product (shown in Appendix A) was eluted (Wizard SV Gel and PCR clean-up system, Promega, Walldorf, Germany) and subsequently sequenced (Eurofins Genomics, Cologne, Germany). The received nucleotide sequence was initially analyzed via NCBIs nucleotide BLAST against the “ITS from fungi type and reference material” database [38]. The ITS sequence of the WCO-isolated KJ27-7 was aligned to the 100 closest BLAST matches using MAFFT v7.471 [39]. A maximum-likelihood phylogenetic tree was constructed using RAxML v8.2.12 [40] on the CIPRES Science Portal [41]. The tree is rooted with *Candida galis* CBS 8842 and was visualized using Figtree v1.4.4. The ITS sequence of the WCO-isolated KJ27-7 has been deposited at Genbank [42] under the accession number MW485779.

### 2.2. Preparation of WSHL

Dried wheat straw with a relative moisture content of 6.6% was obtained from local farmers in Upper Austria. Before steam explosion, the straw was chopped to a particle size of 2–3 cm with an electrical garden shredder (GE260, Viking, Kufstein, Austria). The dry weight of the WS (also steam exploded WS) was determined using an IR moisture analyzer (Ohaus MB45). Steam explosion was performed in a lab scale reactor (VAM GmbH & Co KG, Linz, Austria) as previously described [43,44]. Briefly, 450 g of dried and chopped WS (6.6% moisture content) was mixed 1:1 with dH_2_O, and steam explosion was performed at 200 °C for 10 min at 1.5 MPa. Subsequently, the moisture content of steam exploded WS was determined using an IR moisture analyzer (Ohaus MB45). Subsequently, steam exploded WS was hydrolysed at 10% solids loading. Briefly, WS was mixed in a 1:10 (dryweight:buffer) ratio in 0.1 M citric acid buffer and the pH was adjusted to 5.0. Then 0.3 mL (equals 11 FPU) Accellerase 1500 (Genencor) per g dry weight of WS were added. After incubation at 50 °C and shaking at 120 rpm for 72 h, larger particles were removed by filtration in two subsequent steps using 7–12 µm and 2–4 µm filters. The liquid hydrolysate was sterile filtered and stored at 4 °C.

### 2.3. Strains and Media

The second *I. orientalis* strain that was used in this study (ATCC 24210) was obtained by DSMZ—German Collection of Microorganims and Cell Cultures (#3433). It was maintained on YPD agar plates. Growth curves and shake flask fermentations were performed in 100 mL Erlenmeyer flasks filled with 20 mL medium.

Liquid cultures were prepared as indicated, in either YPD, YNB-D, YNB-DX (YNB-D with 13 g L^−1^ xylose), YNB with 15 g L^−1^ xylose (YNB-X) or YNB-HL. YNB-HL media were prepared by adding sterile-filtered WSHL to 10× YNB stock solutions and diluting with dH_2_O to yield indicated percentage (*v*/*v*) of WSHL and a final YNB concentration of 6.7 g L^−1^. Therefore, YNB-HL90 contained only WSHL and 10× YNB-stock. Unless otherwise indicated, liquid cultures were incubated at 26 °C and 150 rpm in an orbital shaker at a starting OD_600_ of 0.1. Yeast growth was evaluated by optical density measurements with a photometer (Dr. Lange ION 500) at 600 nm (OD_600_).

### 2.4. Spot Drop Test

YNB-D medium containing 15 g L^−1^ agar was prepared and autoclaved. Before pouring the liquid medium in petri dishes, acetic acid, formic acid, furfural, hydroxymethylfurfural (HMF), or ethanol were added at the concentrations indicated. As overnight culture, 10 mL YPD was inoculated with *I. orientalis* KJ27-7. The next day, the overnight culture’s OD_600_ was measured and 10 OD (10^8^ cells) were harvested, washed with dH_2_O and resuspended in 1 mL dH_2_O. Subsequently, dilutions of 1:10, 1:100, 1:1000 and 1:10,000 in dH_2_O were prepared in a 96 well plate. Then, 3 µL of each cell suspension were pipetted onto the inhibitor-containing plates with a multichannel pipette. After drying, the plates were incubated at 26 °C for 1 day and the image recorded by FluorChem FC3 system, Bio-Techne, Minneapolis, MN, USA.

### 2.5. Sugar and Inhibitor Quantification by HPLC

To analyze saccharides and inhibitory compounds, 1 mL of culture was taken at the indicated time points, centrifuged (3000 rpm for 5 min) and the supernatant was diluted (1:10 in dH_2_O) an used for quantification by HPLC, using a Jasco HPLC 2000 plus series (Biolab, Vienna, Austria) with an Aminex HPX 87H column at 65 °C. H_2_SO_4_ (c = 5 mM) was used as eluent at an isocratic flow rate of 0.8 mL min^−1^. Sugars were detected with a refractive index and organic acids and inhibitors by a UV detector. Data were analyzed with ChromPass (Version 1.8.6.1, Jasco Europe, Italy). WSHL was analyzed in the same way, but was diluted 1:50. For the preparation of standards, chemicals were obtained from the following suppliers: xylitol from Sigma Aldrich (Steinheim, Germany), glucose, xylose, arabinose, acetic acid and formic acid from Carl Roth (Karlsruhe, Germany), cellulose form Machery-Nagel (Düren, Germany), furfural from Merck (Hohenbrunn, Germany) and HMF from Alfa Aesar (Kandel, Germany). Resulting saccharide and inhibitor yields (g per 100 g WS) were calculated as follows:$$g\;\mathrm{per}\;100\;g\;\mathrm{WS}=\frac{\left[g\right]\;\mathrm{saccharide}\;\mathrm{per}\;L\;\mathrm{as}\;\mathrm{measured}\;\mathrm{by}\;\mathrm{HPLC}*\mathrm{dilution}\;\mathrm{factor}}{\left[\%\right]\;\mathrm{dry}\;\mathrm{mass}\;\mathrm{of}\;\mathrm{WS}*\left[g\right]\;\mathrm{WS}\;\mathrm{loading}\;\mathrm{per}\;L}$$

Hydrolysates were prepared at 10% solids loading, which means that 100 g WS are contained in 1 L hydrolysate. Therefore, resulting yields in g per 100 g raw material equal g L^−1^ hydrolysate.

### 2.6. Calculation of Relative Ethanol Yields

Calculations of ethanol yields relative to the maximum theoretical yield are based on a maximum theoretical ethanol yield of 0.51 g ethanol g^−1^ glucose. The product yield (Yp, in g ethanol g^−1^ glucose) was calculated according to the percentage and batch of WSHL (see Appendix A) that has been used for the respective cultivation media, by using the following formula:$$g\;\mathrm{ethanol}\;\mathrm{per}\;g\;\mathrm{glucose}=\;\frac{\left[g\right]\;\mathrm{ethanol}\;\mathrm{per}\;L\;\mathrm{as}\;\mathrm{measured}\;\mathrm{by}\;\mathrm{HPLC}}{\left[g\right]\;\mathrm{glucose}\;\mathrm{per}\;L\;\mathrm{in}\;\mathrm{WSHL}\;\mathrm{as}\;\mathrm{measured}\;\mathrm{by}\;\mathrm{HPLC}*\frac{\left[\%\right]\;\mathrm{WSHL}\;\mathrm{in}\;\mathrm{media}}{100}}$$

The relative ethanol yields (in relation to the theoretical maximum) were calculated according to the following formula:(1)$$\;\left[\%\right]\;\mathrm{relative}\;\mathrm{ethanol}\;\mathrm{yield}=\frac{g\;\mathrm{Ethanol}\;\mathrm{per}\;g\;\mathrm{glucose}}{0.51}*\;100$$

### 2.7. Statistical Analysis

All experiments were performed at least three times, each time in triplicates. Statistical analysis was performed with Prism v8.0.2—GraphPad Software, San Diego, CA, USA. Outliers were detected by a Grubbs outlier test and statistical significance was calculated via one-way ANOVA followed by a Fishers LSD test. Values are displayed as means with standard deviation of the mean, *p* < 0.05 = *, *p* < 0.005 = ** and *p* < 0.005 = ***.

## 3. Results

### 3.1. Isolation and Identification of I. orientalis from Waste Cooking Oil

To find yeasts, which are able to grow on unusual substrates and which can be employed for biotechnological applications, microorganisms were isolated from waste cooking oil (WCO). First, 100 µL WCO samples were streaked out on Rose Bengal Red agar plates containing chloramphenicol (RBC). The WCO sample incubated RBC plate from which the strain originated, is provided in Appendix A. Then pure cultures were established from microorganisms that morphologically resembled yeasts. After isolating pure cultures, the microorganisms were identified on a molecular level by analyzing the ITS regions between the small (18S) and large (28S) subunit rDNA. PCR amplification of the respective WCO-isolated strain’s gDNA with ITS5 and ITS4 primers resulted in a product with approximately 550 bp, see Appendix A. A BLAST analysis of the sequenced PCR amplified fragment revealed that the reference sequence with the highest BLAST alignment score was the *I. orientalis* type strain ATCC 6258 (*Pichia kudriavzevii*), with a similarity of 99.15% over 95% of the sequence length. The alignment on the sequences are shown in Figure 1a. Phylogenetic analysis (Figure 1b) further supports the high sequence similarity between WCO-isolated KJ27-7 and *I. orientalis,* as the sequence clusters within a clade of *Pichia* species. Appendix A shows a representative photographic image of WCO-isolated *I. orientalis* KJ27-7 pure culture on YNB-D agar.

### 3.2. I. orientalis KJ27-7 Grows in a Broad Temperature Range

Thermotolerant *I. orientalis* strains, that grow well at temperatures up to 42 °C, have previously been described [7]. Therefore, we analyzed the growth of *I. orientalis* KJ27-7 at various temperatures. As can be seen in Figure 2, the strain grows well in a temperature range from 20 °C to 42 °C. Its optimal growth temperature lies between 20 °C (OD_600_ = 10.9 ± 0.9 after 72 h) and 26 °C (OD_600_ = 11.5 ± 0.5 after 72 h). Although higher temperatures clearly impede the strain’s growth capabilities, it is still able to grow considerably well at 37 °C (OD_600_ = 9.9 ± 0.3 after 72 h) and 42 °C (OD_600_ = 7.9 ± 0.2 after 72 h). Hence, KJ27-7 can be described as thermotolerant.

### 3.3. I. orientalis KJ27-7 Is Tolerant to Acetic Acid, Furfural, HMF and Ethanol

Hydrolysates derived from steam exploded lignocellulosic material such as wheat straw contain several substances that are generally considered as inhibitors of microbial growth and fermentation, among them acetic acid, formic acid, HMF and furfural [45]. It has been described that *I. orientalis* can be fairly tolerant to many of these classical inhibitory substances [7,46]. To evaluate *I. orientalis* KJ27-7’s capability to tolerate the inhibitory substances, which are typically present in steam explosion-derived lignocellulosic hydrolysates, a spot drop test was performed. Serially diluted yeast cell suspensions were pipetted on agar plates containing various concentrations of the indicated substances. As shown in Figure 3, *I. orientalis* KJ27-7 is very tolerant to most of the substances tested (up to 75 mM acetic acid, 5 mM furfural, and 10 mM HMF). However, one substance to which the strain is quite sensitive to is formic acid. KJ27-7 tolerates 10 mM formic acid, but already at 25 mM, a substantial, significant inhibition of growth can be observed. Moreover, also KJ27-7’s tolerance to ethanol was analyzed by this assay. As can be seen in Figure 3, *I. orientalis* KJ27-7 is also relatively tolerant to ethanol. While the strain’s growth is almost unhampered by 10% (*v*/*v*) ethanol, a significant inhibition of growth can only be observed at 15% (*v*/*v*) ethanol. Yet, when incubated for one more day, substantial growth can be observed even in 15% ethanol (Appendix A).

### 3.4. Growth of I. orientalis KJ27-7 on WSHL Media

To investigate the suitability of WSHL as fermentation substrate for *I. orientalis* KJ27-7, first its capability to grow in steam exploded WSHL media was evaluated. The growth in YNB media containing 60, 70, 80 or 90% WSHL as sole carbon source (YNB-HL60, YNB-HL70, YNB-HL80 and YNB-HL90) was observed. As shown in Figure 4, *I. orientalis* KJ27-7 grew very well in WSHL containing media, even better than in the YNB-DX control medium (OD_600_ = 11.8 ± 1.2 after 72 h). While KJ27-7 grew comparably well in YNB-HL60 (OD_600_ = 20.5 ± 1.5 after 72 h), YNB-HL70 (OD_600_ = 21.6 ± 0.9 after 72 h) and YNB-HL80 (OD_600_ = 20.0 ± 1.8 after 72 h), its growth was slightly diminished in YNB-HL90 (OD_600_ = 17.9 ± 0.9 after 72 h).

### 3.5. Ethanol Production of I. orientalis KJ27-7 from WSHL

Since *I. orientalis* KJ27-7 tolerated low double-digit concentrations of ethanol and grew particularly well in WSHL-containing media, next its fermentative capacity to produce ethanol from WSHL-derived sugars was analyzed by HPLC. As performed in the growth analysis experiments, WSHL was used as the sole carbon source for fermentations in shake flasks. As shown in Table 1 and Figure 5, the absolute as well as product yield (Yp) increased with the percentage of WSHL present in the media. In 60% WSHL media (YNB-HL60), *I. orientalis* KJ27-7 produced 5.0 ± 0.6 g L^−1^ ethanol within 24 h, corresponding to a volumetric productivity of 0.21 g L^−1^ h^−1^ and a Yp of 0.37 g ethanol per g glucose (72% of the maximum theoretical ethanol yield). In 70% WSHL media (YNB-HL70), 6.8 ± 0.9 g L^−1^ ethanol were produced in the same time. This corresponds to a volumetric productivity of 0.28 g L^−1^ h^−1^ and a Yp of 0.43 g g^−1^ (83% of the theoretical maximum). In 80% WSHL media (YNB-HL80), *I. orientalis* KJ27-7 produced 8.7 ± 0.5 g L^−1^ ethanol, corresponding to a volumetric productivity of 0.36 g L^−1^ h^−1^ and a Yp of 0.48 g g^−1^ glucose (94% of the theoretical maximum). In 90% WSHL media (YNB-HL90), *I. orientalis* KJ27-7 produced 10.3 ± 1.2 g L^−1^ ethanol within 24 h, corresponding to a volumetric productivity of 0.43 g L^−1^ h^−1^ and a Yp of 0.50 g g^−1^ glucose (97% of the theoretical maximum).

## 4. Discussion

Ethanol production is a widespread trait across many *Pichia* strains. In this study, we isolated a novel yeast strain from WCO that is closely related to *I. orientalis*, which is a member of the *Pichia* clade, as supported by BLAST identity and phylogenetic placement, see Figure 1.

As demonstrated in Figure 2, the strain we isolated is thermotolerant and grows considerably well between 20 °C and 42 °C. [47]. Another *I. orientalis* strain, isolated from corn stalk, also showed a very similar tolerability towards higher temperatures and grew well between 30–42 °C. This strain even grew considerably well at 45 °C [21]. Likewise, *P. kudriavzevii* RZ8-1, which was isolated from soil and decaying fruits, was capable of growing comparably well between 30–42 °C. While only a slight growth limitation of this *I. orientalis* strain could be observed at 42 °C, a clear growth inhibition could be observed starting from 45 °C [7]. This temperature-insensitive growth characteristic of various *I. orientalis* strains that have been isolated from different environmental sources make this organism a promising candidate for a broad variety of process conditions and biorefinery applications. Especially for the production of cellulosic ethanol, which has been a highly investigated subject in recent decades. However, many approaches were facing the problem of the engaged microorganisms’ susceptibility to inhibitory substances derived from lignocellulose pretreatment. *I. orientalis* KJ27-7 is very tolerant to a majority of traditionally renowned steam explosion-derived inhibitory substances. It is particularly tolerant to acetic acid, which is one of the major inhibitory substances of steam exploded lignocellulosic substrates [48]. Figure 3 shows that *I. orientalis* KJ27-7 tolerates up to 75 mM acetic acid, while it hardly grows at 100 mM acetic acid. Similarly, a clear growth inhibition of the thermotolerant *I. orientalis* strain isolated by Chamnipa et al. can be seen at acetic acid concentrations of 75 mM and higher [7]. Formic acid, which is another lignocellulose-derived inhibitor, severely affected *I. orientalis* KJ27-7′s growth at concentrations as low as 25 mM (Figure 3). As demonstrated in Figure 3, a complete inhibition of growth was observed at 50 mM and higher. Likewise growth of *I. orientalis* LC375240, another thermotolerant strain isolated by Ndubuisi et al., was inhibited by 30 mM formic acid [46], whereas the strain isolated by Dandi et al., was completely blocked at 2 g L^−1^ (44 mM) [49]. Moreover, LC375240 tolerated up to 20 mM furfural, whereas the growth of *I. orientalis* KJ27-7 was clearly inhibited by 10 mM furfural. Additionally, a slight but clear inhibition of growth could already be observed at 1 g L^−1^ (10 mM) furfural for the strain isolated by Kwon et al. [21]. While this strain’s growth was markedly decreased already at 1 g L^−1^ (8 mM) HMF [21], *I. orientalis* KJ27-7 showed only minor growth inhibition in 10 mM HMF. Growth of *I. orientalis* KJ27-7 was severely hampered at 25 mM HMF (Figure 3). Overall, the susceptibilities of *I. orientalis* KJ27-7 are comparable to previously isolated *I. orientalis* strains. However, each individual strain tolerates slightly different concentrations of classical lignocellulose-derived fermentation inhibitors and thereby provides certain advantages and disadvantages when applied in biorefinery processes.

Clearly, another key determinant for efficient cellulosic ethanol production is the strains’ tolerance to elevated ethanol concentrations. It has been described that some *I. orientalis* strains are able to be recovered after 48 h incubation in 13% ethanol [50]. *I. orientalis* KJ27-7 grew almost unhampered in 10% ethanol, and a distinct growth inhibition was observed only at 15% ethanol (Figure 3). While some thermotolerant *I. orientalis* strains exhibited a very similar ethanol tolerance profile [46,51], others show a clear growth limitation in the presence of 10% [52] or even 8% ethanol [7].

The fermentation of wheat straw by *I. orientalis* has never been described so far. Hence, we tested various concentrations of WSHL as growth substrate and sole carbon source for *I. orientalis* KJ27-7. Compared to YNB-DX, the strain grows better in YNB media containing WSHL (Figure 4). YNB-DX can be regarded as minimal medium, while WSHL media most probably contain additional salts and other compounds that seem to favor yeast growth. As shown in Figure 4, higher concentrations of WSHL did not inhibit growth of *I. orientalis* KJ27-7. Only slight growth constraints could be observed in the highest WSHL concentration (90%). No growth inhibition owing to furfural or HMF were expected, since only about 0.5 mM (0.05 g L^−1^) furfural and 1.5 mM (0.17 g L^−1^) HMF are present in the WSHL used for media preparation (Appendix A). The observed limitations might be due to formic acid, which is present at 0.17 g L^−1^ (equals 3.5 mM) in 90% WSHL media. Nonetheless, ethanol productivity was highest in 90% WSHL media (Figure 5). Generally, it seems reasonable that the absolute amount of ethanol was elevated in higher WSHL concentrations due to the higher amount of fermentable saccharides present. Nonetheless, higher concentrations of WSHL also mean higher concentrations of potentially inhibitory substances. As discussed above, these obviously do not negatively influence growth, nor the fermentation process. In contrast, the Yp increased in higher WSHL concentrations (72% of the theoretical maximum in YNB-HL60 vs. 97% of the theoretical maximum in YNB-HL90), which demonstrates that augmented WSHL concentrations even favor the specific conversion of glucose into ethanol. The only fermentable carbon sources contained in YNB-HL90 media are about 20 g L^−1^ glucose and 9 g L^−1^ xylose (can be calculated from Appendix A). *I. orientalis* KJ27-7 cannot grow on xylose as sole carbon source (Appendix A) and does not consume xylose, neither in media containing xylose as the sole carbon source (Appendix A) nor the presence of glucose (Appendix A). Therefore, a maximum specific conversion rate of 0.50 g ethanol per g lignocellulose-derived glucose (corresponds to 97% Yp) is calculated for the 90% WSHL medium. Similar values have been reported for *I. orientalis* fermentation of pre-treated corn stalk (94%) [21] or hydrolysates of plywood chips and sugarcane bagasse (91%) [22].

## 5. Conclusions

Generally, *I. orientalis* is a very robust yeast species that is particularly tolerant to various stress factors. Yet, there are inter-strain variabilities, which in turn offer several advantages that can be used for fine-tuning biorefinery processes. Depending on the raw materials used, their pre-treatment and the desired products, individual strains offer distinct features that allow the use of microorganisms tailored to the desired application. *I. orientalis* KJ27-7 is a robust and efficient producer of lignocellulosic ethanol from WS as a low-cost substrate.

## Figures and Tables

**Figure 1 jof-07-00121-f001:**
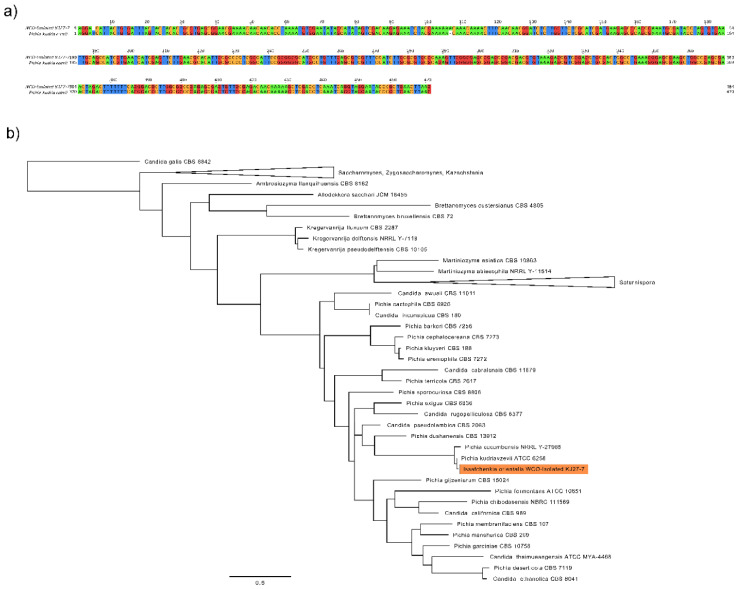
Isolation of *I. orientalis* from WCO. (**a**) Pairwise sequence alignment of WCO-isolated *I. orientalis* KJ27-7 and *Pichia kudriavzevii* ATCC 6258. Alignment of 470 bp region of the ITS region of the genome. Nucleotides are colored (A = green, G = red, T = blue, C = yellow) to display mismatching sites. (**b**) Maximum likelihood phylogenetic tree of WCO-isolated *I. orientalis* KJ27-7 and closest representative sequences. ITS sequence of WCO isolate is highlighted in orange within a clade of *Pichia* spp. The tree is rooted with *Candida galis*.

**Figure 2 jof-07-00121-f002:**
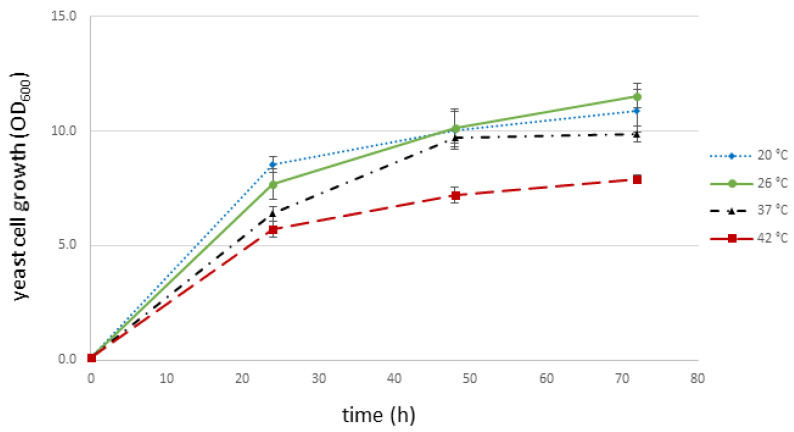
Growth curve of *I. orientalis* KJ27-7 at different temperatures. OD_600_ measurement of *I. orientalis* KJ27-7 in YNB-D media, which contains 20 g L^−1^ glucose incubated at 20 °C (υ), 26 °C (λ), 37 °C (π), or 42 °C (ν) after 24, 48 and 72 h. Displayed are mean values ± SD of four independent experiments each performed in duplicates.

**Figure 3 jof-07-00121-f003:**
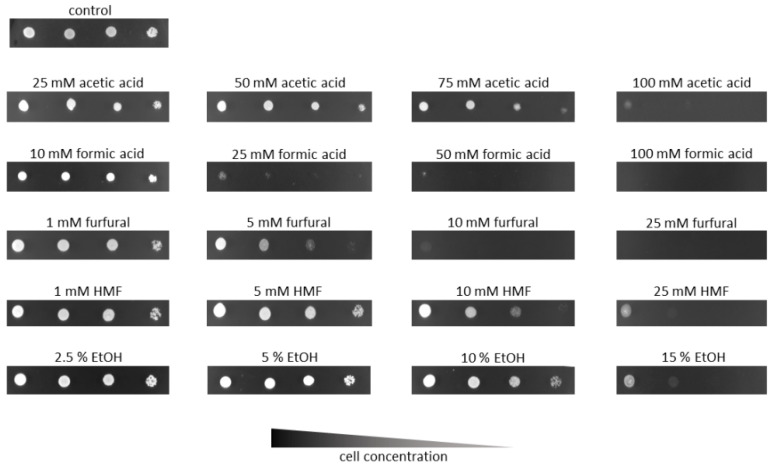
Inhibitor tolerance of *I. orientalis* KJ27-7. Serial dilutions (1:10, 1:100, 1:1000 and 1:10,000) of an *I.* orientalis KJ27-7 cell suspension (10^8^ cells per ml) were spotted on YNB agar plates containing 20 g L^−1^ glucose and the indicated concentrations of either acetic acid, formic acid, furfural, HMF or ethanol (EtOH). Images are taken after 48 h incubation. One out of two representative experiments performed in quadruplicates are shown.

**Figure 4 jof-07-00121-f004:**
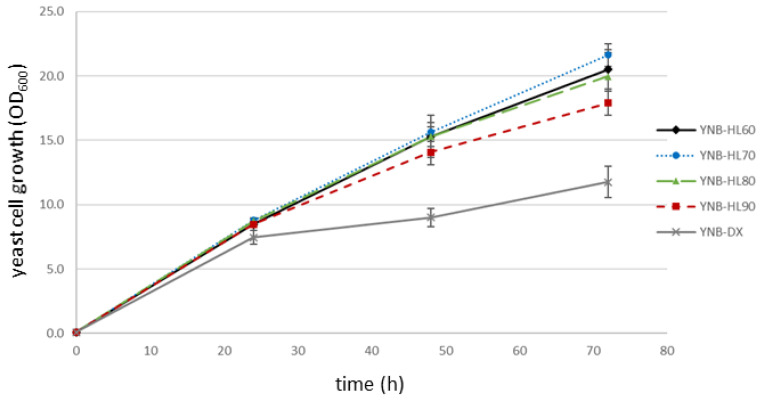
Growth of *I. orientalis* KJ27-7 in WSHL. OD_600_ of *I. orientalis* KJ27-7 in control media with 20 g L^−1^ glucose and 20 g L^−1^ xylose (x, YNB-DX), or in YNB media with 60% (υ, YNB-HL60), 70% (λ, YNB-HL70), 80% (π, YNB-HL80), or 90% (ν, YNB-HL90) WSHL as sole carbon source after 24, 48 and 72 h at 26 °C. Displayed are mean values ± SD of four independent experiments each performed in duplicates.

**Figure 5 jof-07-00121-f005:**
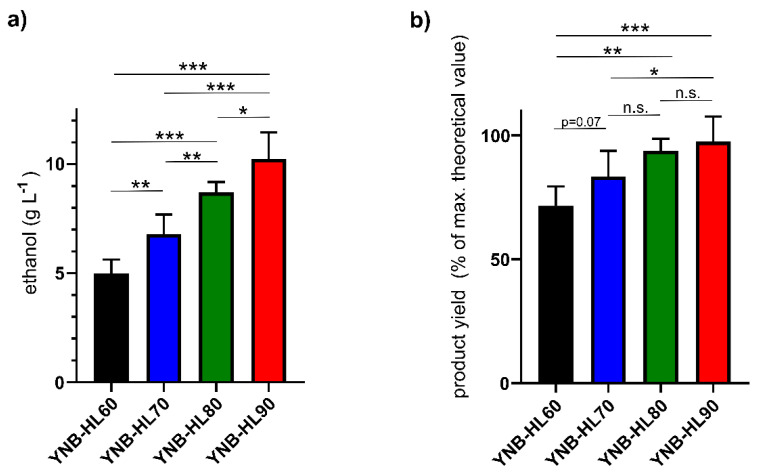
Ethanol production of *I. orientalis* KJ27-7 from WSHL. Ethanol yield (as determined by HPLC) of *I. orientalis* KJ27-7 after 24 h cultivation at 26 °C in YNB media containing 60% (black), 70% (blue), 80% (green) or 90% (red) WSHL as the sole carbon source. (**a**) absolute values in g L^−1^; (**b**) product yield (Yp) in percent of maximum theoretical ethanol yield. Displayed are mean values ± SD of four independent experiments, each performed in duplicates. Asterisks indicate *p*-values (** = *p* < 0.005, * = *p* < 0.05 and *** = *p* < 0.001).

**Table 1 jof-07-00121-t001:** Ethanol Production of *I. orientalis* KJ27-7.

	Total Ethanol after 24h (g L^−1^)	Volumetric Productivity (g L^−1^ h^−1^)	Product Yield (Yp) (g g^−1^ Glucose)
YNB-HL60	5.0 ± 0.6	0.21	0.37 ± 0.04
YNB-HL70	6.8 ± 0.9	0.28	0.43 ± 0.06
YNB-HL80	8.7 ± 0.5	0.36	0.48 ± 0.02
YNB-HL90	10.3 ± 1.2	0.43	0.50 ± 0.05

## Data Availability

The data presented in this study are openly available in GenBank, reference number MW485779.

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
