# Peer review of "Ethanol Production from Wheat Straw Hydrolysate by Issatchenkia Orientalis Isolated from Waste Cooking Oil"

_jof, 2021, doi:10.3390/jof7020121_

Round 1

Reviewer 1 Report

Zwirzitz and coworkers report on the isolation of a novel Issatchenkia orientalis KJ27-7 strain that has been isolated of waste cooking oil. The strain is an oleaginous in nature and thus has relatively high thermo- and chemotolerancy. The authors provide a clear and standard description on their sampling, colonies growth, obtaining of pure culture, isolation of the genomic and PCR amplification of the internal transcribed spacer. The strain has been well fitted in the phylogenetic tree.

Using HPLC techniques the authors have followed the production of ethanol wheat straw hydrolysate derived sugars. The obtained results are in accordance with the claim that the new strain has a relatively high ethanol production.

Overall, the work is well written, rhetorical and scientifically coherent. A minor remark is that the authors should place labels on the axis in Figure 2 and Figure 4.

Author Response

The positive comments of reviewer 1 are highly appreciated. Also the minor remark was gratefully acknowledged. The axis of Figure 2 and Figure 4 have been changed accordingly.

Reviewer 2 Report

Authors demonstrated the possibility of the effective conversion wheat straw hydrolysate into ethanol by a novel strain Issatchenkia orientalis isolated from waste cooking oil. The manuscript is well written and easy to read. The presented manuscript can be published in the Journal of Fungi, however the corrections to the manuscript are required.

Point 1: Page 1, lines 23-24 – please to change “a specific productivity” for “product yield (Yp)”.

Point 2: Page 2, lines 60-65 - The purpose of the study must be reformulated in traditional manner.

Point 3: Page 4, lines 158-160 - please to change “a specific productivity” for “product yield (Yp)”.

Point 4: Page 4, line 175-176– please to change “In guest for yeast” for “To find yeasts”

Point 5: Page 8, Table 1 - please to change “a specific productivity” for “product yield (Yp)”.

Point 6: Table 1 and Figure 5 (a) present the same data. It is better to put data from Figure 5 (b) into Table 1.

Author Response

Reviewer 2's highly appreciated comments are very constructive and thus gladly considered in the revised version of the manuscript. See below for the point-by-point response.

Point 1: Page 1, lines 23-24 – The term “a specific productivity” has been changed to “product yield” as suggested.

Point 2: Page 2, lines 60-61 - The purpose of the study is now described in a traditional way, as suggested. Therefore, a sentence has been introduced in lines 60-61.

Point 3: Page 4, lines 158-160 - The term “a specific productivity” has been changed to “product yield (Yp)” as suggested.

Point 4: Page 4, line 175-176– The phrase “In quest for yeasts" has been changed to “To find yeasts”, as suggested.

Point 5: Page 8, Table 1 - The term “a specific productivity” has been changed to “Product yield (Yp)” as suggested.

Point 6: We certainly agree that Table 1 and Figure 5a present the same data. However, in addition to the data, Figure 5 also shows the statistic evaluation thereof. This additional information would be lost if the data were included in Table 1.  Moreover, we feel that the visual display of the data adds additional value. Therefore, we hope that reviewer 2 understands our argument and accepts Table 1 and Figure 5, as shown in the manuscript. Nonetheless, if requested, we are of course open to omit Figure 5 and include the data from Figure 5b into Table 1.